# Automatic generation of object shapes with desired functionalities

## Abstract

3D objects (artefacts) are made to fulfill functions. Designing an object often starts with defining a list of functionalities that it should provide, also known as *functional requirements*. Today, the design of 3D object models is still a slow and largely artisanal activity, with few Computer-Aided Design (CAD) tools existing to aid the exploration of the design solution space. The purpose of the study is to explore the possibility of shape generation conditioned on desired functionalities. To accelerate the design process, we introduce an algorithm for generating object shapes with desired functionalities. We follow the principle *form follows function*, and assume that the form of a structure is correlated to its function. First, we use an artificial neural network to learn a function-to-form mapping by analysing a dataset of objects labeled with their functionalities. Then, we combine forms providing one or more desired functions, generating an object shape that is expected to provide all of them. Finally, we verify in simulation whether the generated object possesses the desired functionalities, by defining and executing functionality tests on it.

## 1 Motivation

Design cycles of products are lengthy, as they usually involve thousands of decisions on the form of the product that will implement the desired functionalities. Despite efforts in the last two decades to accelerate the workflow using CAD techniques (Kurtoglu, 2007; Autodesk, Inc.), most of the design process is still done manually. In an attempt to solve this pertinent problem, the Defense Advanced Research Projects Agency (DARPA) launched in 2017 the *Fundamental Design* call for research projects on conceptual design of mechanical systems, that would enable the generation of novel design configurations DARPA (2017). The purpose of our study is to explore the possibility of automatic shape generation conditioned on desired functionalities, as illustrated in Fig. 1.

Figure 1: Generated object shape providing the *containability* and *supportability* functionalities.

This paper also has another motivation stemming from robotics. Traditionally, research in autonomous robots deals with the problem of recognising affordances of objects in the environment: i.e. given an object, what actions does is afford to do? Given a shape, what are its functionalities? This paper addresses the inverse problem: given a list of functionalities, what shape would provide all of them?

This paper presents a method and an architecture for automatic generation of object shapes with desired functionalities. It does so by autonomously learning mappings from object form to function, and then applies this knowledge to conceive new object forms that satisfy given functional requirements. In a sense, this method performs *functionality arithmetic* (by analogy with *shape arithmetic* (Wu et al., 2016)) through manipulation of latent vectors corresponding to functionalities (as opposed to shapes). Fig. 2 illustrates the concept: combine *features* describing two different objects to create another object possessing the *functionalities* of both initial objects. A quick skim through the other figures of this paper may help the reader understand what we are talking about. A second contribution is the use of experiments to verify the presence of affordances in the generated object shapes using a physics simulator, both by defining explicit tests in the simulator, and by using state-of-art affordance

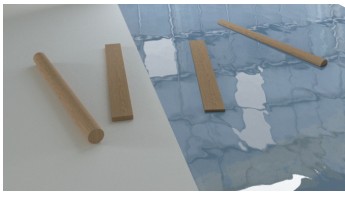
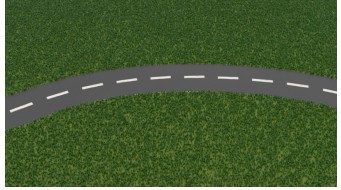
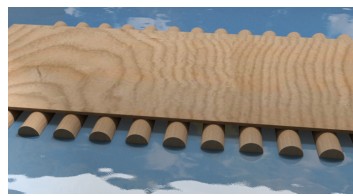

(a) *Wooden* beams have the *float-ability* affordance.

(b) *Flat* roads have the *traverse-ability* affordance.

(c) *Flat wooden* roads (pontoon bridges) offer both *float-ability* and *traverse-ability*.

Figure 2: The features that describe (2a) wooden beams and (2b) flat roads can be combined, to obtain an object design that possesses both *float-ability* and *traverse-ability*: (2c) a pontoon bridge.

detectors. To summarise, this paper has three contributions: (i) a novel method for extracting and combining functional forms of object shapes, (ii) design and execution of specific validation tests for the presence of desired affordances, (iii) a novel network architecture specialised for modeling 3D shapes.

This remainder of the paper is organised as follows. Section 2 presents an overview of the related work in object design, shape descriptors, and object affordances. Section 3 describes our methodology, detailing the envisioned workflow for using this technology, and details regarding the architecture of the network. It also describes the operators employed for object form manipulation. In Section 4 we discuss the obtained results and describe the drawbacks of the method at its current state. Finally, in Section 5 we draw a conclusion and detail the opportunities for future work. In this paper we will use the terms *affordance* and *functionality* interchangeably.

## 2 RELATED LITERATURE

The literature review is organised in three sections, detailing the state-of-the-art in the three fields at the intersection of which this study finds itself: object design, object shape descriptors (for manipulation of object forms), and learning of object affordances (for relating object forms to functionalities).

### 2.1 OBJECT DESIGN

The idea of getting inspiration from previous designs when conceiving a new object is not new, and appears under names such as Analogical reasoning, and Design reuse. A standard practice in design is to consult *knowledge ontologies* (Bryant et al., 2005; Kurtoglu & Campbell, 2009; Bhatt et al., 2012) that contain function-to-form mappings (Umeda & Tomiyama, 1997; Kurtoglu, 2007). However, the knowledge acquisition required to populate such ontologies involves a (non-automated) process known as *functional decomposition*, in which a human analyses existing objects by disassembling them into components and noting the functionality provided by each component. A related review on object functionality inference from shape information is presented in (Hu et al., 2018).

Recently, generative design emerged as an automated technique for exploring the space of 3D object shapes (Autodesk, Inc.) using genetic algorithms. It formulates the shape search as an optimisation problem, requiring an initial solution, a definition of parameters to optimise, and rules for exploring the search space. However, it is far from trivial to identify rules for the intelligent exploration of the shape space, that would provide results in reasonable time. In a similar context of generative design, Umetani (2017) employed an AutoEncoder to explore the space of car shapes.

### 2.2 OBJECT SHAPE DESCRIPTORS

Object shape descriptions serve two purposes: (1) they contain extracted object shape features, which are used to study the form-to-function relationship, and (2) they serve as basis for the reconstruction of 3D object models. State-of-the-art techniques for automatically extracting object features are practically all based on Neural Networks, typically Convolutional Neural Networks or Auto-Encoders

(Girdhar et al., 2016), which have replaced the methods based on hand-crafted features like Scale-invariant feature transform (SIFT) or Speeded up robust features (SURF).

In order to generate 3D shapes from descriptions, modern techniques also employ Neural Network approaches: Auto-Encoders (Girdhar et al., 2016) and Generative Adversarial Networks (Wu et al., 2016), which learn a mapping from a low-dimensional probabilistic latent space to the space of 3D objects, allowing to explore the 3D object manifold. In this study, we used a Variational AutoEncoder (VAE) (Kingma & Welling, 2013; Rezende et al., 2014) to both extract features describing 3D objects, and reconstruct their 3D shapes when given such a description.

## 2.3 OBJECT FUNCTIONALITIES AS OBJECT AFFORDANCES

A field of research that also focuses on linking objects with their functionalities is that of *affordance learning*. It is based on the notion of *affordance* that defines an action that an object provides (or affords) to an agent (Gibson, 1977). In the context of this paper, we are interested in approaches that map object features to corresponding object affordances (or functionalities). A common approach is to extract image regions (from RGB-D frames) with specific properties and tag them with corresponding affordance labels. An overview of machine learning approaches for detecting affordances of tools in 3D visual data is available in (Ciocodeica, 2016). Recent reviews on affordances in machine learning for cognitive robotics include (Jamone et al., 2016; Min et al., 2016; Zech et al., 2017).

This paper introduces a method to automatically learn shape descriptors and extract a form-to-function mapping, which is then employed to generate new objects with desired functionalities. The novelty lies in the use of what we call *functionality arithmetic* (operations on object functionalities) through manipulation of corresponding forms in a feature space. This is an application of the principle *form follows function* in an automated design setting. Following this principle, we assume that object forms are correlated to their function. Moreover, since we extract shape features from a dataset of objects designed by humans for humans, it is reasonable to assume that their shapes are close to optimal for performing their intended function.

## 3 METHODOLOGY

The purpose of the study is to explore the possibility of shape generation conditioned on desired functionalities. The main idea is to train a VAE to generate voxel occupancy grids, and then generate novel shapes by combining latent codes from existing examples with desired functionalities.

The working hypotheses are: (i) objects providing the same functionality have common form/shape features, (ii) averaging over multiple shapes that provide the same functionality will extract a form providing that functionality, that we call "functional form", (iii) parametric interpolation between samples can generate novel shapes providing the combined functionalities of those samples. This last assumption is contentious, as we cannot yet predict the behaviour of functionalities when combining their underlying shapes. For this reason, we verify the presence of these functionalities in simulation.

We employed a voxelgrid representation for 3D object models as it satisfied the requirement of convolutional AutoEncoders to have fixed-size input, and was the easiest to employ for a proof-of-concept.

The starting point for this research was the hypothesis that object functionalities arise due to features that those objects possess. Therefore, if we intend to create an object with a desired set of functionalities, then it should possess corresponding features providing these functionalities.

In this section we describe our workflow for object generation (Section 3.1), and the *functionality arithmetics* operators that we used for generating shapes with desired functionalities (Section 3.2). Technical details on the employed neural network architecture and its training are available in Section A.1 of the appendix.

## 3.1 PROPOSED WORKFLOW

For employing the proposed object generation method, we suggest a workflow composed of two phases: (1) learning phase, in which a neural network is trained to generate feature-based representations of objects and to faithfully reconstruct objects using this representation, and (2) request phase, in which a user requests the generation of a novel object with some desired functionalities among those present in the traning dataset of affordance-labeled objects. The algorithm would then pick object categories providing those functionalities, extract the shape features responsible for providing those functionalities (generating the form-to-function mapping), and combine them to generate a feature description of a new object. This description would then be used to generate a 3D model of the desired object.

## 3.2 OPERATORS

In this section we describe the operators that we employed for manipulating object forms. Section 3.2.1 will describe the extraction of *functional form* of a category of objects, which is the set of features that provides the functionalities of that category. Section 3.2.2 will describe how we combine two object descriptions into a single new one, that is expected to have the functionalities of both input objects.

### 3.2.1 EXTRACT THE *functional form* OF A CATEGORY OF OBJECTS

Every category of objects possesses a set of functionalities that defines it. From a *form follows function* perspective, all objects samples contained in a category share a set of features that provide its set of functionalities. We call this set of features the *functional form* of a category of objects. Multiple methods may exist for extracting it. For example, Larsen et al. (2015) isolated face features (e.g. presence of glasses, bangs, mustache) by computing the difference between the mean vector for categories with the attribute and the mean vector for categories without the attribute. In our particular case, we compute the vector of shape features that are responsible for the presence of functionalities as the average latent vector of an object category. This functional form of an object category can then be visualised by inputting the obtained feature (latent-vector) description into the decoder trained to reconstruct 3D volumes. Fig. 4 illustrates some results obtained using this method.

Later, we assign an importance value to each latent variable composing the *functional form* of a category. We do this by computing the Kullback–Leibler (KL) divergence between the Probability Density Function (PDF) of these variables with the PDF of (1) variables describing a void volume, and (2) a non-informative distribution of independent Gaussians with 0 mean and unit variance (called *prior*). The motivation behind using these two KL divergences for ranking the variables is to identify (1) which variables make the shape different from a void volume, to capture the filled voxels of the model, and (2) which variables distinguish the shape description from that of a Gaussian prior. Both of these KL divergences are normalised, so as to have unit norm. Then, an *importance vector* is defined as the weighted sum of the normalized KL divergences with a void and a non-informative prior distribution, with the corresponding weights $w_{void} = 1/2$ and $w_{prior} = 1/2$ chosen empirically.

### 3.2.2 COMBINE THE *functional forms* OF TWO DIFFERENT CATEGORIES OF OBJECTS

In order to combine two object descriptions (i.e. two latent vectors containing these descriptions), we need to identify which of the variables in each vector are important for encoding the object shape. In a degenerate case, if all the variables are critical for encoding the object shape, then their values cannot be changed, and therefore the object cannot be combined with another one (or a conflict resolution function must be devised). The hypothesis is that not all the variables are critical for representing the object shape, meaning that some variables' values can be neglected when combining two object descriptions. We identify which variables are important for an object description using the *importance vector* method described above in Section 3.2.1.

The combination of two object descriptions is guided by their corresponding *importance vectors*. For simplicity, we describe the combination as being made between two object descriptions, although the method is applicable to any number of objects. One object serves as a *base object*, from which are taken the initial values of the latent variables' distributions for the *combined object* description. The

Table 1: Interaction cases between latent variables contained in the descriptions of two different objects ($Obj_{base}$, $Obj_{top}$), which appear when attempting to combine them.

| # | Latent variable from $Obj_{base}$ | Latent variable from $Obj_{top}$ | Latent variable from $Obj_{combined}$ |
|---|---|---|---|
| 1 | non-important | non-important | value of base object |
| 2 | non-important | important | value of important variable |
| 3 | important | non-important | value of important variable |
| 4 | important | important | average of the two values |

other object serves as *top object*, whose latent variables' distributions are combined with those of the *base object* according to the rules described in Table 1. The degree to which two object categories are combined can be controlled by varying the amount of information kept from each object description (i.e. the percentage of variables considered *important* for an object description).

Four cases appear when combining two latent vector descriptions of objects, as seen in Table 1. These rules can be resumed as follows: if both variable distributions are important then average them (case 4 in the table), if only one is important then keep the important one (cases 2 and 3 in the table), else keep the base values (case 1 in the table).

Figure 5 shows some outcomes of using these combination rules, including the impact of the order in which objects are combined, and of different threshold levels for the importance vectors (50%, 60%, 70%, 80%, and 95%). The impact of different combination parameters on the output 3D models is shown in Fig. 11 of the Appendix.

Employing a void volume as *base object* onto which the important features were overlaid did not result in satisfactory outputs, as the models were mostly void. However, using a void volume as base would have made the combination function commutative.

## 4 RESULTS AND DISCUSSION

In this section we provide our results on the (a) capacity of the VAE to describe and reconstruct objects, (b) extraction of functional forms for different categories of objects, (c) generation of novel objects through the combination of feature representations of object categories containing desired functionalities, and (d) affordance testing for the generated objects. At the end of this section, we discuss the limitations of the proposed method.

### 4.1 OBJECT REPRESENTATION AND RECONSTRUCTION RESULTS

Fig. 3 illustrates 3D object samples and their corresponding reconstructions generated by the network. The satisfactory quality of reconstructions suggests that the encoder network can generate descriptions of objects in a feature (latent vector) space, and that the decoder network can successfully reconstruct objects from descriptions generated by the encoder network.

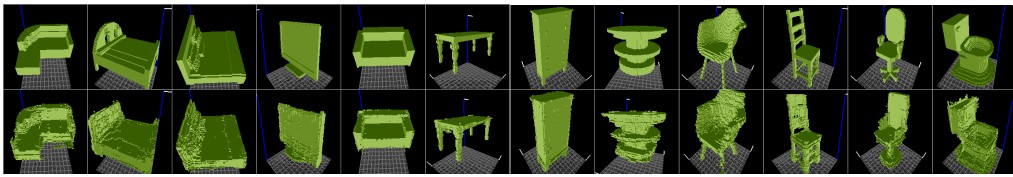

Figure 3: Examples of original voxelised objects (top) and their reconstructions (bottom) generated by the VAE neural network. Objects taken from the ModelNet dataset (Wu et al., 2015).

### 4.2 FUNCTIONAL FORM EXTRACTION RESULTS

Through the extraction of functional forms of different object categories, we expected to identify forms that provide functionalities offered by those categories of objects. Fig. 4 shows results on

*functional form* extraction for tables, chairs, and monitors. Relevant features have been extracted, such as the flatness of tables providing *support-ability*, the seats and backrests of chairs providing *sit-ability* and *lean-ability*, respectively. In the case of the chair object category, a considerable proportion of objects had armrests, which led to this feature becoming part of the functional form.

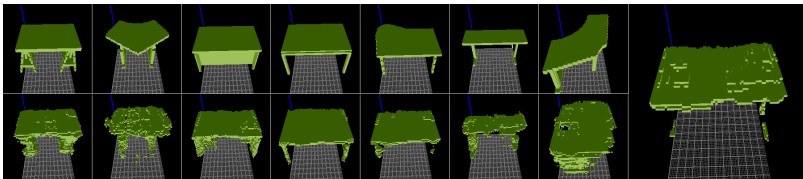

(a) Sample tables (top), their reconstructions (bottom), and their common form features or *functional form* (right). The flatness feature was successfully extracted, which can be interpreted as providing the *support-ability* of tables. Since supports differed in the samples, their were not included in the set of common shape features.

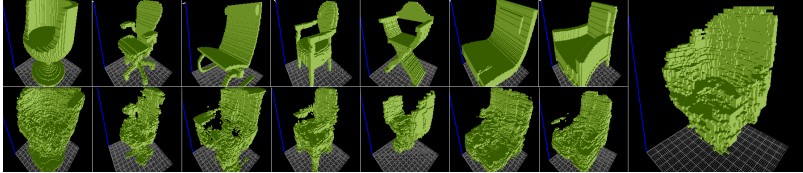

(b) Sample chairs (top), their reconstructions (bottom), and their common form features (right). The seat and backrest are present in the set of common shape features, providing the *sit-ability* and *lean-ability* affordances. Since multiple chairs had armrests in the samples, these were also included in the set of common shape features.

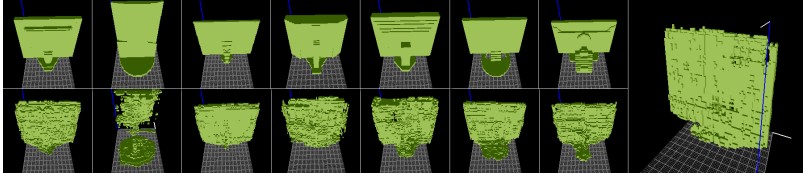

(c) Sample monitors (top), their reconstructions (bottom), and their common form features (right). The flatness of screens was successfully identified as a common shape feature.

Figure 4: **Functional forms** extracted for (a) tables, (b) chairs and (c) monitors. Objects taken from the ModelNet dataset (Wu et al., 2015). Visualiser: viewvox (Min, 2004).

## 4.3 OBJECT COMBINATION RESULTS

The ability to extract a shape representation that constitutes the functional form of a category, coupled with the ability to combine it with another object representation, makes it possible to extract and combine shape features that provide desired functionalities. It is worth noting that the proposed combination operator is non-commutative, meaning that the combination of two objects can generate different results, depending on the order of objects in the combining operation (i.e. which object is used as *base object*, and the order in which other objects are combined with it).

### 4.3.1 SIT-ABILITY AND WASH-ABILITY

In this experiment, we have attempted to extract the *sit-ability* and *wash-ability* of toilet seats and bathtubs, respectively, in order to combine them into a new object providing both of these functionalities. The obtained results may be interpreted as bidet objects.

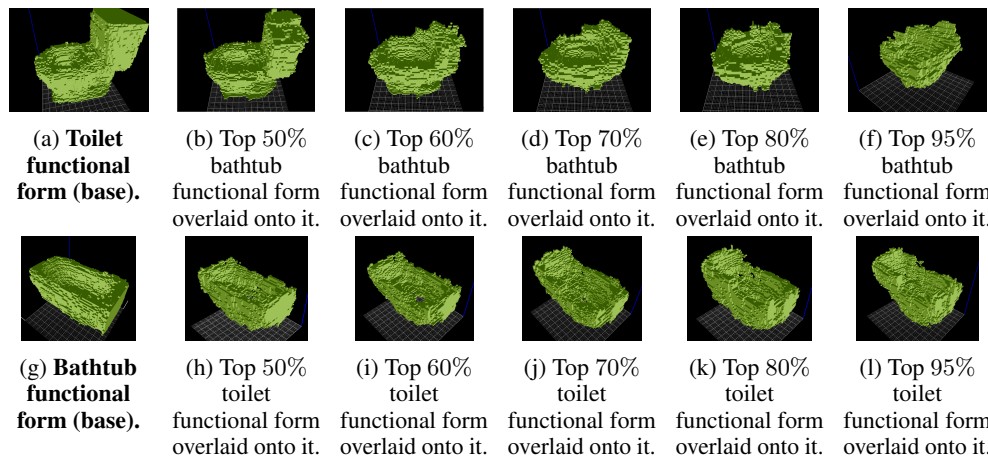

(a) **Toilet functional form (base).**

(b) Top 50% bathtub functional form overlaid onto it.

(c) Top 60% bathtub functional form overlaid onto it.

(d) Top 70% bathtub functional form overlaid onto it.

(e) Top 80% bathtub functional form overlaid onto it.

(f) Top 95% bathtub functional form overlaid onto it.

(g) **Bathtub functional form (base).**

(h) Top 50% toilet functional form overlaid onto it.

(i) Top 60% toilet functional form overlaid onto it.

(j) Top 70% toilet functional form overlaid onto it.

(k) Top 80% toilet functional form overlaid onto it.

(l) Top 95% toilet functional form overlaid onto it.

Figure 5: **Object combination** results for bathtubs and toilets functional forms, using a **toilet functional form base combined with a bathtub functional form** (top), and a **bathtub functional form base combined with a toilet functional form** (bottom), both of which can be interpreted as a bidet. A gradual transformation is displayed (base functional form combined with top-50% to top-95% of the second functional form). From left to right, the combination looks less like a toilet (top) / bathtub (bottom) and more like a bidet.

### 4.3.2 WASH-ABILITY AND SUPPORT-ABILITY

This experiment displays the combination of *support-ability* and *contain-ability* functionalities with the intent of creating something similar to a workdesk in a bathtub. The result is shown in Fig. 6.

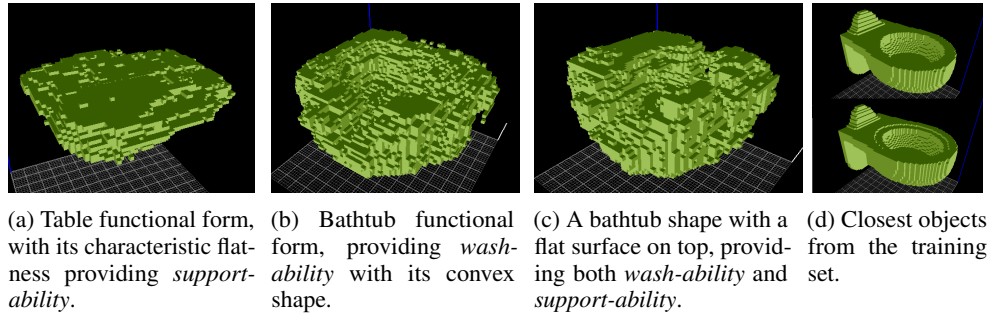

(a) Table functional form, with its characteristic flatness providing *support-ability*.

(b) Bathtub functional form, providing *wash-ability* with its convex shape.

(c) A bathtub shape with a flat surface on top, providing both *wash-ability* and *support-ability*.

(d) Closest objects from the training set.

Figure 6: Combining features of objects providing respectively *wash-ability* and *support-ability* into a novel object form, providing both functionalities. (6d) shows the two objects from the training dataset that are closest to our generated object, in terms of similarity of the activation values of the one-before-last layer of the decoder.

### 4.4 QUANTITATIVE RESULTS

We analysed the generated objects using 3 methods: (1) verification of affordance presence using state-of-art affordance detectors, (2) comparison of generated objects to most similar objects in the dataset, and (3) testing affordance presence in a physics simulation. These are detailed below.

### 4.4.1 AFFORDANCE DETECTORS

We attempted to identify the presence of the desired affordances (contain-ability, support-ability) using affordance detectors developed by other groups.

Sadly, we were not able to replicate the affordance detection results of Myers et al. (2015) on synthetic object images seen by a Kinect RGBD camera inside the Gazebo simulator. It failed to recognise the containability affordance in both standard objects like a bowl and a saucepan, and in generated ones.

We also tried the affordance detector of Do et al. (2018), called AffordanceNet. While it worked on objects viewed in simulation (including those of objects from the ModelNet40 dataset on which our network was trained), it had difficulties with recognising properly the affordances of generated objects (see Fig. 7a). We found experimentally that the failure cases for affordance detection were caused by the rugged surface of the object, and the fact that AffordanceNet was not trained on images of rugged objects. After applying Poisson smoothing to this object's surface, the detector correctly identified the presence of *contain-ability*, although it still struggled to locate it properly (see Fig. 7b).

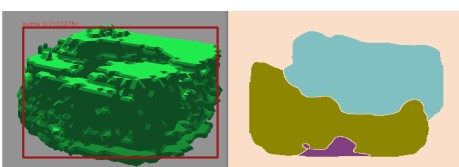 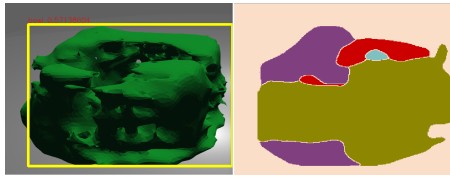

(a) The AffordanceNet detector correctly identified *support-ability* (in light blue) and *wrap-grasp-ability* (in mustard colour), and incorrectly identified *hit-ability* (in purple).

(b) On a smoothed version of the object, and in different lighting conditions, it correctly identified *wrap-grasp-ability* (mustard), *contain-ability* (red), although with imperfect segmentation. It incorrectly identified *hit-ability* (purple) and *support-ability* (light blue).

Figure 7: Affordance detection results using the AffordanceNet (Do et al., 2018).

### 4.4.2    MOST SIMILAR SHAPES IN THE TRAINING DATASET TO THE ONES GENERATED

To ensure that the employed algorithm does not simply generate models by copying samples from the dataset, we compare the generated objects with the most similar samples from the dataset, based on the similarity of outputs of the one-before-last layer of the decoder. The result from Fig. 6d confirmed that generated objects are distinct from the samples in the training set.

### 4.4.3    AFFORDANCE TESTING IN SIMULATION

To verify that the generated objects indeed provide the requested affordances, we developed some tests to execute in simulation. For this purpose, the generated voxelgrid model is transformed into a mesh using the *marching cubes* method (Lorensen & Cline, 1987), after which we compute its inertia matrix and create the Spatial Data File (SDF) file that allows to import it into the Gazebo simulator, using the Bullet physics engine.

To verify for supportability, we suspended the object into the air, and verified which of its regions can support a stable object with a flat base, by dropping from above from different (x,y) locations a 0.1 $m^3$ cube with mass 1 kg, and checking whether this had any impact on the (x,y) coordinates of its centroid. If only its z coordinate (altitude from ground) had changed, while the (x,y) coordinates remained the same, then that location was marked as providing stability. On the contrary, if the region was not flat, the cube would tumble over, landing on (x,y,z) coordinates distinct from its initial ones. Fig. 8c shows the obtained result. To verify containability, we dropped spheres into the object until they overflowed, and measured the ratio of the total volume of all spheres contained inside the object versus the volume of its bounding box (see Fig. 8d).

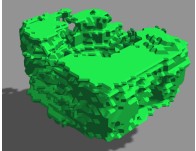 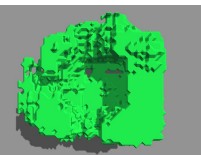 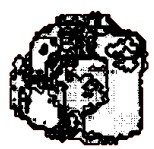 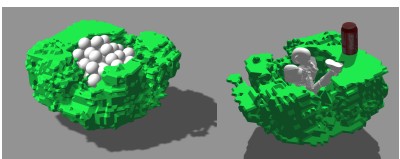

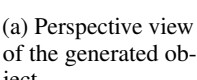

(a) Perspective view of the generated object.

(b) Top-down view of the generated object.

(c) White pixels show locations with supportability.

(d) Containability test results with spheres.

(e) Containability of a humanoid robot in the bathtub.

Figure 8: The generated bathtub-workdesk object in (8a) perspective view and (8b) top-down view. (8c) shows the result of the *support-ability* test, while (8d) shows the result of the *contain-ability* test. (8e) demonstrates that an iCub humanoid robot can fit inside the bathtub, and the Coca-Cola can illustrates the supportability of the workdesk.

## 4.5 LIMITATIONS

The proposed method currently has a set of limitations: (i) The method used for extracting functional forms from object categories, which employs *averaging* out the gaussians describing the voxel locations, requires all samples in the dataset to be aligned. (ii) The combination method does not state if a solution to the posed problem does not exist (i.e. if combining two different sets of affordances is possible). (iii) The different scales of objects are not taken into consideration when combining objects. Training the neural network on object models which are correctly sized relative to each other would solve this issue. However, it would require increasing the size of the input voxel cube to fit inside detailed descriptions of both small-scale objects (e.g. spoons, forks, chairs) and large scale objects (e.g. dressers, sofas), which would also increase the training time. (iv) Since the features describe the voxels mostly in the center of the bounding cube, combining two different feature descriptions makes them compete for the same center voxels in this bounding volume. Introducing an operator for spatially offseting some shape features would allow to construct composite objects. For instance, if we want to extract the *sit-ability* and *support-ability* from chairs and tables, respectively, in order to create something similar to a conference chair, it would be required to offset the table features with respect to the chair features.

## 5 CONCLUSION AND FUTURE WORK

We have presented a method for generating objects with desired functionalities, by first extracting a form-to-function mapping from a dataset of objects, and then manipulating and combining these forms through functionality arithmetic. The method relies on a neural network to extract feature-based descriptions of objects. These descriptions allow shape manipulation and arithmetics in a latent feature space, before being transformed back into 3D object models. We then test the presence of desired affordances in a physical simulator, and with an affordance detector.

In contrast to an ontology based approach, where modifications can be done deterministically, all the object shape manipulations are probabilistic in our case. Thus, generated inexact models may prove sufficient if regarded only as *design suggestions*. However, a production-grade technology would require less noisy object-modeling results. We plan to employ a Generative Adversarial Network (GAN) approach (Wu et al., 2016), encouraging the network to generate objects with smooth surfaces similar to those of existing man-made objects. We also plan to implement a training procedure to encourage neurons in the latent layer to represent specific transformations (rotation, scale) following the approach of Kulkarni et al. (2015).

Our models still lack information about materials from which objects are composed, their colors or textures (where necessary), and the articulations between subparts. Adding them would make the approach much more practical.

The voxelgrid representation comes with a tradeoff (easy to use, but low model resolution, high computational complexity for training the neural network), and it is possible to revise this decision in our future work. This would require considering alternative representations such as point clouds, mesh representations, shape primitives, etc.

In addition, instead of using a dataset containing an implicit mapping of form-to-function (as objects are categorised in categories), we intend to learn object functionalities/affordances automatically, by letting a robot interact autonomously with a set of objects. This is related to the currently active field of *affordance learning* in robotics. Moreover, the use of 3D shape descriptors developed in this research will facilitate affordance learning and knowledge transfer in the case of autonomous robots.

The source code will be made available upon publication.

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

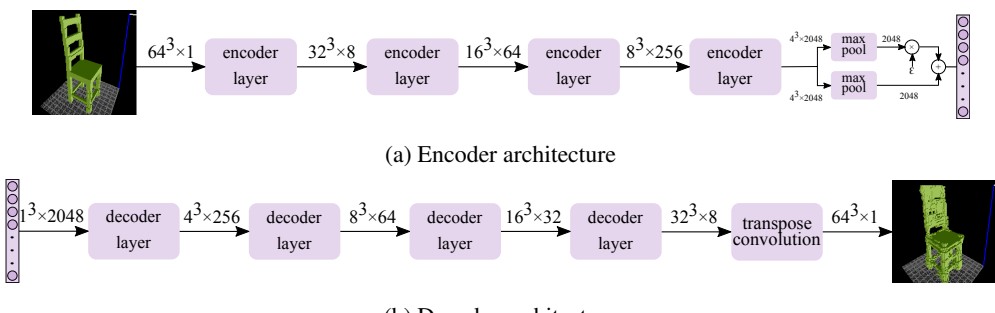

(a) Encoder architecture

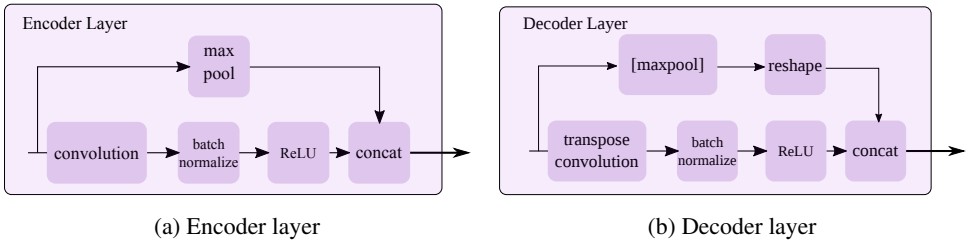

(b) Decoder architecture

Figure 9: The architectures of the (a) encoder and (b) decoder.

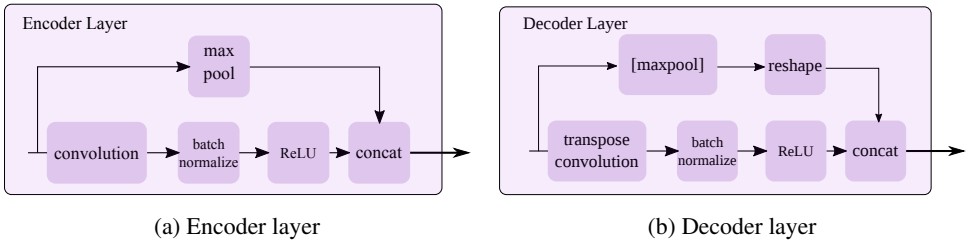

(a) Encoder layer                    (b) Decoder layer

Figure 10: Schematics of the building blocks of the (a) encoder layer and (b) decoder layer.

## A  APPENDIX

### A.1  NEURAL NETWORK ARCHITECTURE

To come up with an automatic method for describing the features of objects, we employ a VAE that we train on the ModelNet 40 dataset of common household objects (Wu et al., 2015). This dataset contains 3D models of bathtubs, beds, chairs, desks, dressers, monitors, night stands, sofas, tables, toilets, etc. (see examples in Fig. 3). For processing, we convert the samples in the dataset from OFF to binary voxelgrid (BINVOX) format (Min, 2004; Nooruddin & Turk, 2003), obtaining exact voxelgrid models centered in a volume of dimension 64x64x64 voxels. We augment the dataset by rotating the voxelgrid models by 90, 180 and 270 degrees around their vertical axis.

The network architecture is shown in Figure 9. The inputs are cubes of size $64 \times 64 \times 64$, which is identical to the dimensions of the reconstructed outputs. Both the encoder and the decoder employ convolutional layers, interspersed with rectified linear unit (ReLU) non-linearities and batch normalization (Ioffe & Szegedy, 2015) operations. Inspired by DenseNet architecture (Huang et al., 2017), we have stacked the outputs of activation layers throughout the encoder and decoder layers. However, unlike the bottleneck layers of (Huang et al., 2017), we have used max-pool operations (and reshape operation, in the case of the decoder) to align the shapes (see Figure 10). This was motivated by the fact that a significant bottleneck already exists in the latent variable layer.

The last layer of the encoder performs a reduce-max operation, in order to generate the means and variances for the gaussian distributions that model each of the variables of the latent vector. The VAE employs a latent vector of size $2^{11}$ (i.e. 2048) latent variables, which serves both as a bottleneck and as container of the object description. We use a VAE loss to train the network, which is composed of two parts: weighted binary cross-entropy (the reconstruction loss) and KL divergence with a non-informative prior (a Gaussian with zero mean and unit variance) which is a regularisation loss. For a single example, the (non-weighted) reconstruction loss is computed as follows:

$$\sum -\big(x \cdot log(x') + (1 - x) \cdot log(1 - x')\big) \tag{1}$$

where $x$ is the data, and $x'$ is the reconstruction. To improve training speed, we employ a weighted cost function, that penalises proportionately more the network for errors in reconstructing full voxels

than for errors in reconstructing empty voxels:

$$\mathbb{E}_{q(z|x)}[log\ p(x|z)] = \sum -\big(\alpha \cdot x \cdot log(x') + (1 - x) \cdot log(1 - x')\big) \tag{2}$$

where $\alpha$ is the weight factor for the filled voxels, $log(.)$ is applied element-wise, and the summation is over the whole volume. This is useful, since on average most of the reconstructed volume is empty, while the objects occupy only $\approx 4\%$ of all the voxels. This allows to avoid the local minimum trap at the beginning of training, when the network prefers to reconstruct only empty volumes. We empirically set this weight to a value of $\alpha = 10$. The regularisation loss is computed as:

$$D_{KL}(q(z|x)||p(z)) = \frac{1}{2}\sum_{j=1}^{J}(1 + log(\sigma_j^2) - \mu_j^2 - \sigma_j^2) \tag{3}$$

where $J$ is the number of latent variables ($2048 = 2^{11}$ in our case), and $\mu_j$ and $\sigma_j$ are the parameters of the posterior of the latent variables (i.e. probability of the latent vector given the observation of a single data point: $q(z|x)$). To improve the quality of reconstructions, we use a modified version of Eq. 3, referred to as *beta VAE* (Higgins et al., 2017) that encourages the network to use all the capacity of its latent layer:

$$\mathbb{E}_{q(z|x)}[log\ p(x|z)] - \beta D_{KL}(q(z|x)||p(z)) \tag{4}$$

with $\beta = 0.01$ kept constant during training.

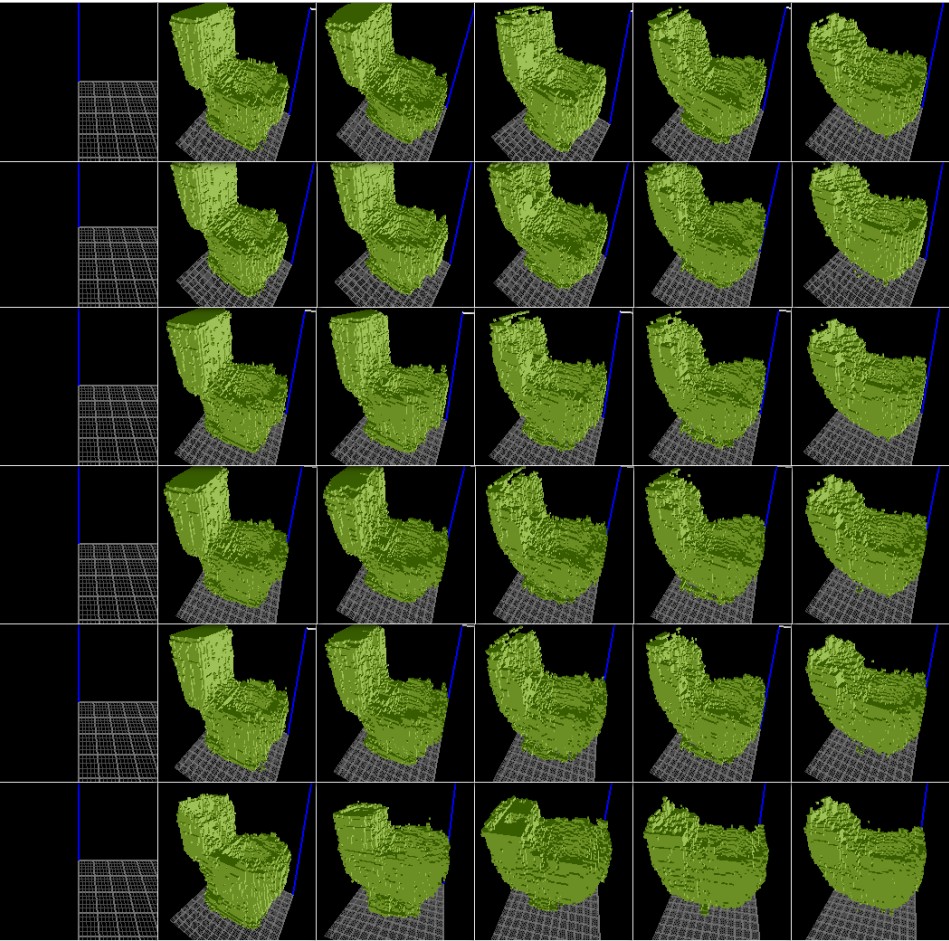

Figure 11: Impact of the different parameters used when combining shape encodings. **Left to right:** amount of total variables used (ranging from 0 to 1, in steps of 0.2), ranked by their importance score. The left-most column has only void, as no variables are used for combinations. The right-most column corresponds to interpolations between the functional forms of a toilet and a bathtub. **Top to bottom:** weight of KL divergence with the encoding of a void volume (from 0 to 1, in steps of 0.2); the weight of KL divergence with a Gaussian prior is (1 - KL divergence with void).

