# OpenReview forum: "Automatic generation of object shapes with desired functionalities"
_ICLR.cc/2019/Conference_

### Official Review · AnonReviewer1 · 2018-11-02
**Cool idea, but not well written, and not sufficiently evaluated**

**Rating:** 3
**Confidence:** 4

**Review:**

This paper presents a method for generating 3D objects. They train a VAE to generate voxel occupancy grids. Then, they allow a user to generate novel shapes using the learned model by combining latent codes from existing examples.

Pros:
- The idea of linking affordances to 3D object generation is interesting, and relevant to the machine learning and computer vision communities.

- They propose to evaluate the quality of the shape based on a physical simulation (Section 4.4.3), which is an interesting idea.

Cons:
- This paper is not well written. The method is described in too much detail, and the extra length (10 pages) is unnecessary. Cross entropy, VAEs, and many of the CNN details can usually just be cited, instead of being described to the reader.

- The paper uses suggestive terminology, like "functional essence" and "functional arithmetic" for concepts that are fairly mundane (see Lipton and Steinhardt, 2018 for an extended discussion of this issue). For example, the "functional essence" of a class is essentially an average of the VAE latent vectors (Section 3.3.1). The paper claims, without sufficient explanation, that this is computation is motivated by the idea that "form follows function".

- The results are not very impressive. There is no rigorous evaluation. They propose several nice metrics to use (eg. affordance simulation), but the results they present for each metric are quite limited. The qualitative results are also not particularly compelling.

- The paper should more thoroughly evaluate the importance weighting that is described in Section 3.3.2.

 - The technical approach (combining VAE vectors to make new shapes) is not particularly novel[

Overall:

The paper should not be accepted in its current form, both due to the confusing writing, and the lack of careful evaluation.

---

> ### Public Comment · (anonymous) · 2018-11-26
> **Thank you for the review**
>
> Thank you for the time you took to review our paper.
> We appreciate the reviewer's insight in summarising our paper and addressing the main points.
>
> Following your remarks, we have reorganised the article, extruding and placing the part about the neural network architecture into an appendix.
>
> Regarding the first concern on the structure of the paper, the initial purpose of describing the details of the architecture was to make the paper self-sufficient. However, according with the feedback we received from the community, we agree with the reviewer that some details are verbose. The part on the neural network architecture was removed from the main body of the paper, and left for an optional appendix.
>
> Regarding the second concern on the employed terminology, we re-read the paper by Lipton and Steinhardt, and renamed the term "functional essence" into "functional form" of an object.
> We preferred this term, as it denotes the purpose of this operation.
> We also did not name it "averaging of latent vectors" because there  may be multiple methods for extracting this "functional form" (one is presented by us, another one from [2] is cited).
> We thus preferred to use the term "functional form extraction" for a family of algorithms performing this task.
>
> The term "functional arithmetic" makes use of the analogy with the term "shape arithmetic" [1]. Following this parallel, we manipulate latent vectors corresponding to _functionalities_ (as opposed to _shapes_ in the cited reference). Thus, we argue to maintain this term.
>
> Regarding the computations being motivated by the idea "form follows function":
> We follow the principle form follows function, and assume that the form of an object is correlated to its function. Moreover, since we extract shape features from a dataset of objects designed by humans for humans, it is reasonable to assume that the employed shapes are close to optimal for performing their intended function. We included this mention in the paper.
>
> Regarding the third concern on the results:
> We would like to point out that (to the best of our knowledge) there are no alternative methods for shape generation conditioned on desired functionalities.
> Hence, it would be misleading to state that they are not at the level of the state-of-the-art.
>
> As this research is based on exploring new concepts, detailed quality of the reconstructions is not the major contribution of our work. Rather, we try to formulate a new problem for generating shapes based on functionalities. For instance, the generated bathtub-workdesk does provide the desired functionalities, but its aesthetics should improve with further research.
>
> Regarding the evaluation of the importance weighting described in Section 3.3.2, we added images of the combination of toilet and bathtub functional forms, to show the interpolation spectrum (see Fig. 11 in the Appendix).
> We admit that a rigorous evaluation would have required an evaluation of all the shapes generated using different combination parameters, in order to choose the parameters that consistently provide best results. This will become meaningful once we will have a bigger set of affordance/functionality tests.
> For the moment, we decided to view this multitude of solutions as design proposals, leaving the final choice for the human designer.
> Hopefully, these new additions brought the paper closer to the desired rigour standard.
>
> Regarding the remaining concerns:
> Indeed, the employed architecture is not conceptually novel (it is still a 3D autoencoder). However, the same paper by Lipton and Steinhardt [3] mentioned above also states that "empirical advances often come about [...] through clever problem formulations [...] or by applying existing methods to interesting new tasks." We consider the formulation of the problem of shape design conditioned on desired functionalities/affordances valuable in itself.
>
> To summarise, we used the advice from the reviewers and revised accordingly the paper (changes are highlighted in yellow). We hope that the quality of the writing has improved.
>
> Thank you again for the time invested into this review.
>
> References:
> [1] Jiajun Wu, Chengkai Zhang, Tianfan Xue, Bill Freeman, and Josh Tenenbaum. Learning a probabilistic latent space of object shapes via 3d generative-adversarial modeling.
> In D. D. Lee, M. Sugiyama, U. V. Luxburg, I. Guyon, and R. Garnett (eds.), Advances in Neural Information Processing Systems 29, pp. 82–90. Curran Associates, Inc., 2016.
>
> [2] Larsen, Anders Boesen Lindbo, et al. "Autoencoding beyond pixels using a learned similarity metric." arXiv preprint arXiv:1512.09300 (2015).
>
> [3] Lipton, Zachary C., and Jacob Steinhardt. "Troubling trends in machine learning scholarship." arXiv preprint arXiv:1807.03341 (2018).

---

### Official Review · AnonReviewer2 · 2018-11-03
**An ad-hoc method for shape generation**

**Rating:** 3
**Confidence:** 4

**Review:**


This paper proposed a 3D shape generation model. The model is essentially an auto-encoder. The authors explored a new way of interpolation among encoded latent vectors, and drew connections to object functionality.

The paper is, unfortunately, clearly below the bar of ICLR in many ways. It’s technically incremental: the paper doesn’t propose a new model; it instead suggests new way of interpolating the latent vectors for shape generation. The incremental technical innovation is not well-motivated or justified, either: the definitions of new concepts such as ‘functional essence’ and ‘importance vector’ are ad-hoc. The results are poor, much worse compared with the state-of-the-art shape synthesis methods. The writing and organization can also be improved. For example, the main idea should be emphasized first in the method section, and the detailed network architecture can be saved for a separate subsection or supplementary material.

It’s good that the authors are looking into the direction of modeling shape functionality. This is an importance area that is currently less explored. I suggest the authors look into the rich literature of geometry modeling in the computer graphics and vision community, and improve the paper by drawing inspiration from the latest progress there.

---

> ### Public Comment · (anonymous) · 2018-11-26
> **Thank you for the review**
>
> Thank you for the time you took to review our paper.
> The summary represents well the contributions of our paper.
>
> Regarding the statement about the incremental contribution of the paper, we would like to emphasise once more that the main contribution of the paper is the formulation of the shape design problem conditioned on desired functionalities/affordances.
> To the best of our knowledge, this problem formulation is novel, and no other method exists which could compete with our method to solve the problem of generating 3D shapes based on functionality requirements.
> In this respect, we believe the problem formulation is novel and cannot be considered an "incremental contribution".
>
> Regarding the fact that the paper does not propose a new model:
> Indeed, the employed architecture is not conceptually novel (it is still a 3D autoencoder). However, Lipton and Steinhardt [3] state that "empirical advances often come about [...] through clever problem formulations [...] or by applying existing methods to interesting new tasks." We consider the formulation of the problem of shape design conditioned on desired functionalities/affordances valuable in itself.
>
> Regarding the "ad-hoc" definition of new concepts:
> We renamed the term "functional essence" into "functional form" of an object.
> We preferred to keep this term, as it denotes the purpose of this operation.
>
> We believe multiple methods would be capable to extract the "functional form" of an object category, of which the "averaging of the latent vector" is only one method, which has served reasonably well in the particular case of our study. To prove this point, we cite another method with a similar purpose, employed in a context of extracting facial features [2].
> We thus preferred to use the term "functional form extraction" for the task that can be performed by potentially an entire family of algorithms.
> To prevent future readers from falling into the same pitfall, we have explained our reasoning behind the introduction of this vocabulary in the paper.
>
> Regarding the "ad-hoc" aspect of the importance vector:
> The problem of deciding which features are important in an object description is raised when two object descriptions have to be combined into a single new one. This motivates the use of the term "importance vector".
>
> Regarding the evaluation of the importance weighting described in Section 3.3.2, we added images of the combination of toilet and bathtub functional forms, to show the interpolation spectrum (see Fig. 11 in the Appendix).
>
> The motivation behind using these two KL divergences for ranking the variables is to identify (1) which variables make the shape different from a void volume, to capture the filled voxels of the model, and (2) which variables distinguish the shape description from that of a Gaussian prior. This mention was added to the manuscript.
>
> Regarding results:
> We bring to the attention of the reviewer that the "normal" aspect (e.g. smooth shapes) of objects was not the purpose of this study, and that the generated objects fulfill their functions as verified by our tests, even though they look rugged. Their shapes are sub-optimal for the selected function (some object parts are useless), but this paper does not claim that they are optimal.
> In this respect, we find that the evaluation of the quality of the generated objects is subjective, and not based on their ability to perform the desired function.
> This being said, the control over the style of object surfaces (smooth vs rugged, plain vs decorated) is an ongoing work, but is not in the scope of this paper.
> It would also be incorrect to state that the generated objects are below the state-of-the-art, as (to our knowledge) no other methods exist for object generation conditioned on desired functionalities.
>
> Regarding the literature review:
> We will certainly look into the latest literature on geometry modeling in the computer graphics and vision community, as these are research areas that considerably overlap with our work. At the same time, if the reviewer is aware of any related literature, we would appreciate it if it could be shared with us.
>
> As suggested by the reviewer, we emphasised the main idea at the beginning of the Methodology section. We also added an image of an automatically generated object on the first page of the paper.
>
> Regarding the quality of writing, (as stated above) we used the advice from the reviewers and revised accordingly the paper (changes are highlighted in yellow). As suggested, we moved the detailed description of the network architecture into the (optional) appendix. We hope that the quality of the writing has improved.
>
> Thank you again for the time invested into this review.
>
> [2] Larsen, Anders Boesen Lindbo, et al. "Autoencoding beyond pixels using a learned similarity metric." arXiv preprint arXiv:1512.09300 (2015).
>
> [3] Lipton, Zachary C., and Jacob Steinhardt. "Troubling trends in machine learning scholarship." arXiv preprint arXiv:1807.03341 (2018).

---

> > ### Comment · AnonReviewer2 · 2018-11-30
> > **Thanks**
> >
> > Thanks for your reply and revision. I keep my original rating.

---

### Official Review · AnonReviewer3 · 2018-11-05
**Interesting work that would deserve a better focus of experiment design**

**Rating:** 5
**Confidence:** 3

**Review:**

This paper is addressing several research challenges as a method to generate objects with desired functionalities, a method to extract form-to-function mapping, a method to operationally support a functionality arithmetic. The work illustrated in this paper is really interesting and is addressing relevant and open problems in the domain of product design.

Nevertheless the manuscript has a couple of weaknesses, one concerned with the presentation and another related to the design of the study.

The lack of a consistent choice for the lexicon is sometimes misleading. It is not always clear whether the use of different terms is addressing synonyms or to discriminate between two distinct concepts. For example let consider the following pairs: functionality versus affordance, function versus functional, class versus category, feature versus shape.

The study addresses several questions. Not always is clear what is the purpose or better the research questions that are driving the design of the experiments. While in the manuscript the are many repetition of the objectives of the study, less attention is devoted to explain what are the working hypothesis underlying the proposed methods. For example, one of the objective is a method to generate objects with desired functionalities. Only in the final Section there is a brief mention of the dichotomy between meash-based versus voxel-based. As reported in Section 2 there are in literature other works but there is not a claim on what is the specific purpose of the present study. The contrast of voxel versus mesh looks like a motivation but it only a speculation. A similar comment might address the dichotomy deterministic (ontology) versus probabilistic (autoencoder). In this case the experiment design should provide some empirical evidence about this contrast.

A minor comment. Figure 7a is illustrating the functional essence of table. According to the caption Figure 5a is illustrating the same functional essence for the same category/class table. Should the pictures look the same?

---

> ### Public Comment · (anonymous) · 2018-11-26
> **Thank you for the review**
>
> Thank you for the time you took to review our paper.
> The summary represents well the contributions of our paper.
>
> Regarding the presentation:
> We added an image of a generated object to the front page of the paper, to show the reader from the beginning what type of models we generate. We also emphasised the main idea at the beginning of the Methodology section.
>
> We agree with the comment regarding the consistent choice of lexicon, and we have purified the employed vocabulary to avoid ambiguities in the meanings of words.
> "Affordance" and "functionality" do not mean the same thing, as affordances also include the actor performing the action, being represented as a tuple (actor, action, object, effect). However, we make the connection between the two concepts, as the literature on the relationship between object features and functionalities can be found using the keywords "affordance learning and recognition".
>
> The paper clearly states that "functionality" and "affordance" are used interchangeably. For all the other ambiguous words, we have improved the manuscript by consistently choosing the same words for the same concepts.
> Fot better comprehension, we have replaced all the instances of the "class" word with "category".
>
> Regarding the "feature" vs "shape" comment:
> Different types of features exist: shape, colour, edges, interest points, etc.
> We focus on shape features, since the network only decides which voxels should be filled/empty. We emphasise this by always mentioning that we work with "shape features".
>
> The purpose of the study is to explore the possibility of shape generation conditioned on the desired functionalities.
> The main working hypotheses are:
> - objects providing the same functionality have common form/shape features
> - averaging over multiple shapes that provide the same functionality will extract a form providing that functionality, that we call "functional form".
> Features that are frequently observed inside an object category will pass the selection threshold to be included in this "functional form". Features that are rarely observed are considered non-relevant for performing the function, and are left out.
> - parametric interpolation between samples can generate novel shapes providing the combined functionalities of those samples.
> This last assumption is contentious, as we cannot yet predict the behaviour of functionalities when combining their underlying shapes.
> For this reason, we verify the presence of these functionalities in simulation.
> We added this mention in the beginning of the "Methodology" section.
>
> Regarding the choice of 3D representation:
> We do not address the question of which 3D representation is better for representing shapes (mesh representations, voxel grids, point clouds, superquadrics, shape primitives, etc.). We chose a voxelgrid representation because it allowed to have fixed-size 3D models as inputs for an autoencoder.
> During functionality testing, the voxelgrid models were converted into mesh models only because the employed simulator (Gazebo) required mesh models.
> This mention was added in the third paragraph of the Methodology section.
>
> Regarding the choice between ontology methods vs probabilistic methods:
> We do not argue against any particular method in this paper.
> We attempt to automate ontology methods (for example [4]), that involved decision making by a human designer on how to combine multiple shapes where each provides some functionality. This shape combination process is automated using a neural network.
>
> Regarding Figures 5a and 7a:
> Figure 5a illustrates the average latent vector over the seven displayed "table" samples.
> Figure 7a illustrates the average latent vector over all the samples inside the "table" category (~400 samples), augmented with their rotations by 90 degrees.
>
> Thank you again for the time invested into this review.
>
> [4] Kurtoglu, Tolga, and Matthew I. Campbell. "Automated synthesis of electromechanical design configurations from empirical analysis of function to form mapping." Journal of Engineering Design 20.1 (2009): 83-104.

---

### Meta-Review · Area_Chair1 · 2018-12-14
**questions regarding usefulness of the problem formulation given non impressive empirical outcomes**

**Confidence:** 5
**Recommendation:** Reject

**Metareview:**

The paper presents a novel problem formulation, that of generating 3D object shapes based on their functionality. They use a dataset of 3d shapes annotated with functionalities to learn a voxel generative network that conditions on the desired functionality to generate a voxel occupancy grid. However, the fact that the results are not very convincing -resulting 3D shapes are very coarse- raises questions regarding the usefulness of the proposed problem formulation.
Thus, the problem formulation novelty alone is not enough for acceptance. Combined with a motivating application to demonstrate the usefulness of the problem formulation and results, would make this paper a much stronger submission. Furthermore, the authors have greatly improved the writing of the manuscript during the discussion phase.